# Understanding Psychological Symptoms of Endometriosis from a Research Domain Criteria Perspective

**DOI:** 10.3390/jcm12124056

**Published:** 2023-06-15

**Authors:** Katharina van Stein, Kathrin Schubert, Beate Ditzen, Cornelia Weise

**Affiliations:** 1Heidelberg University Hospital, Institute for Medical Psychology, 69115 Heidelberg, Germany; 2Faculty of Behavioral and Cultural Studies, Ruprecht Karls-University, 69115 Heidelberg, Germany; 3Department of Psychology, Division of Clinical Psychology and Psychotherapy, Philipps University of Marburg, 35032 Marburg, Germany

**Keywords:** endometriosis, infertility, pelvic pain, RDoC, stress-related diseases, women’s health

## Abstract

Endometriosis is currently the second most common gynecological disease and is associated with severe pain, vegetative impairment, and infertility. In association, there are considerable psychological symptoms that limit the quality of life of those affected. In this narrative review, the Research Domain Criteria (RDoC) framework was utilized to display the different transdiagnostic processes involved in disease progression and maintenance in regard to psychosocial functioning. Using the RDoC framework, it becomes clear that immune/endocrinological dysregulation is interlocked with (pelvic) pain chronification processes and psychological symptoms such as depressive mood, loss of control, higher vigilance toward the onset or worsening of symptoms, social isolation, and catastrophizing. This paper will discuss and identify promising treatment approaches, in addition to medical care, as well as further research implications. Endometriosis can come with substantial psychosomatic and social burden, requiring more research to understand the interdependence of different factors involved in its chronic development pathway. However, it is already clear that standard care should be extended with multifaceted treatments addressing pain, as well as the psychological and social burden, in order to halt the cycle of aggravation of symptoms and to improve quality of life for patients.

## 1. Introduction

Endometriosis is a chronic inflammatory disease that is defined by the growth of endometrial-like tissues outside of the uterus [1]. It is estimated to affect up to 10% of premenopausal individuals with ovaries and a uterus and, although classified as benign, the tissue implants can spread and damage affected organs [2]. The pathogenesis of endometriosis is not clear and studies report a delay from symptom onset to diagnosis of 10 years on average [3].

Endometriosis can be asymptomatic, but is predominantly associated with chronic pelvic pain (CPP), dysmenorrhea, dyspareunia, dysuria, dyschezia, and infertility [2]. Symptom presentation varies across the menstrual cycle and between patients; however, importantly, there is no clear link between pain symptoms and endometriosis stage or localization of tissue implants [4,5,6]. This, and the fact that there is a high prevalence of psychological symptoms in endometriosis patients [7,8], suggests that endometriosis is not exclusively a gynecological condition. 

In addition to somatic symptoms, endometriosis patients frequently experience depressive mood and heightened anxiety [9], higher levels of perceived stress, as well as various kinds of pain [10], all of which influence their social life [11]. Infertility/subfertility and concerns about potential infertility may also lead to worry, depression, and feelings of inadequacy [12]. Cross-sectional studies find higher risks for the diagnosis of depression, generalized anxiety disorder [13], and post-traumatic stress disorder [14] in patients with endometriosis. Previous reviews have illustrated that endometriosis reduces psychosocial wellbeing [15] and overall quality of life (QoL) in patients [16]. Nevertheless, the unclear pathogenesis of endometriosis includes the etiology of its psychological symptoms [17], which is yet to be fully understood.

The Research Domain Criteria (RDoC) are an evolving research structure considering the major domains of neuropsychosocial functioning instead of categorizing symptoms. Promoted by the US National Institute of Mental Health, the RDoC project provides a multidimensional approach to understanding mental health and illness with six key domains, each including several constructs. These constructs can be measured by different units of analysis, ranging from genes to self-reports. The RDoC were not primarily designed for clinical use, but as a research tool to integrate (neuro-)biological findings into the understanding of psychopathology [18]. This transdiagnostic approach, which focuses not only on symptomatic and behavioral aspects, but also includes underlying neurobiological mechanisms, seeks to inspire translational research for the better prevention and treatment of mental illness [19]. The RDoC take the interdependence of psychological symptoms and physiological circuits into account, making it a promising tool for investigating the complex somatopsychic connections in endometriosis.

The aim of this narrative review is to not only demonstrate the burden that patients with endometriosis carry, but to also detect transdiagnostic interrelations in the pathogenesis and perpetuation of psychological symptoms to further identify possible prevention and treatment approaches addressing these mechanisms. Therefore, the review of current evidence was structured by the two RDoC domains of Negative Valence Systems and Systems for Social Processes, since these seem to capture the psychopathology of endometriosis most accurately; the Negative Valence Systems are linked to symptoms of depression [20], anxiety [21], and post-traumatic stress [22], while Systems for Social Processes cover the social factors relevant to endometriosis such as social withdrawal, perceived injustice, and social support [23]. Both will shortly be illustrated at the beginning of each respective section.

With regard to levels of psychosocial functioning, the focus of this review is on units of analysis starting on the level of circuits, going up to physiology, behavior, and self-report. The units of analysis, genes, molecules, and cells, and their relevance regarding psychosocial functioning in endometriosis, were left out of this review and can be found elsewhere [24,25].

The primary literature search was conducted from date of inception (May 2021) until August 2022 via the platforms pubmed, PsycInfo, and PSYNDEX. The initial search term was “endometriosis psych*” and no temporal restriction was used. Subsequent literature searches included topics and authors that were discovered during the initial search.

## 2. Negative Valence Systems

The domain Negative Valence Systems includes adverse motivation and responses to adverse situations or contexts [26]. It is divided into the five subconstructs: Acute Threat, Potential Threat, Sustained Threat, Loss, and Frustrative Nonreward. Acute Threat and Frustrative Nonreward are being left out of this review since they are not considered to be relevant for a comprehensive understanding of the psychological symptoms of endometriosis. 

### 2.1. Potential Threat

Potential Threat refers to an activation due to potential harm that is distant and uncertain or of low certainty [27]. The Generalized Unsafety Theory of Stress [28] postulates that the human stress response serves as a default mode, which can be deactivated through the perception of safety signals. The theory can explain prolonged stress responses in the absence of acute stressors.

Living with endometriosis always entails the possibility of disease progression, of the development of new symptoms, and of the worsening of persisting symptom manifestation [29]. Endometriosis patients live with potential low imminence threats, such as pain or flare-ups, and without remediate treatment, which could otherwise serve as a signal of safety. Therefore, endometriosis is often experienced as a highly uncontrollable disease [30]. Uncontrollable stress led to higher rates of endometriosis progression in rats [31]. The absence of safety signals might play a role as it could possibly lead to higher vigilance in regard to onset or worsening of symptoms. This, in turn, supports constant symptom monitoring as an additional factor for a dysregulated stress response. People with diagnosed endometriosis are also more likely to be diagnosed with post-traumatic stress disorder [14] compared to people without endometriosis. In particular, people who experienced physical or sexual abuse during childhood are more at risk of developing endometriosis later in life. Harris et al. [32] report a 79% higher risk of developing endometriosis for those who experienced both severe physical and sexual abuse during childhood. Childhood abuse and PTSD can leave victims with the concept of the world as an unsafe place in general [33], causing further suspicion of potential threat in many situations of everyday life. Reis et al. [30] conclude that childhood stress, e.g., negligence and abuse, should be considered a risk factor for the development of endometriosis, since these adverse events may cause persistent alterations in the neural and hormonal stress responses [34] relevant to pain severity and disease progression, such as a chronic inflammatory response and dysregulated hypothalamic–pituitary–adrenal axis (HPA axis).

### 2.2. Sustained Threat

The RDoC subconstruct of Sustained Threat describes prolonged exposure to negative experiences, either external or internal. Some patients are exposed to unpleasant states, such as chronic pain and light or heavy bleeding [35], on most days, not only during certain phases of their menstrual cycle. Chronic pain, among other symptoms, often equals chronic stress and contributes to the lasting dysregulation of the HPA axis in individuals with endometriosis [36]. This dysregulation often leads to higher levels of pro-inflammatory agents which lower the pain threshold [37] and, in turn, can cause higher subjective chronic stress [36]. Even the treatment and day-to-day management of the disease and of subfertility is possibly perceived as a sustained threat. Lazzeri et al. [38] found a link between treatment intensity and levels of perceived stress in endometriosis patients with a strong association between repeated surgery and higher self-reported measures of psychological stress. Research on other long-term effects of medical treatment on HRQoL is relatively scarce. Most studies on long-term mental health effects report an overall positive outcome of both pharmacological [39,40] and surgical treatment [41,42]. In their review, D’Alterio et al. [43] report that surgical and pharmacological treatments have comparable long-term effects on pain levels and QoL. However, the follow-up intervals in these studies were rather short (up to 18 months). This is critical, since pain recurrence after surgery can occur many months later [44,45], which might, in turn, lead to a lower QoL.

### 2.3. Loss

The Loss subconstruct of the negative valence systems refers to both the episodic and sustained unwanted disappearance of any object or situation that is not easy to replace. It includes loss of relationships, status, or behavioral control, and is associated with negative emotions as well as rumination and possible shifts in attention. The subjective experience of loss is the result of individual evaluation based on values and beliefs, leading to interindividual differences regarding the extent and intensity of perceived loss. Oftentimes, patients with endometriosis must deal with many kinds of loss from all areas of life: they are likely to lose predictability in everyday life [46], resulting in possible loss of income [10] as well as loss of social relationships, satisfying sex life (see Section 3), and hobbies due to the interference of symptoms with social and other activities [47].

Furthermore, some patients report experiencing loss of their identity as a woman because of possible struggles with fertility and not being able to meet society’s expectation of womanhood [48]. The burden through infertility becomes even higher with experienced pregnancy loss [49]. In their qualitative study, Hållstam et al. [50] summarized living with endometriosis as a constant struggle for coherence with difficulties in establishing meaning and feeling understood. Patients described feelings of loneliness and guilt, sorrow over childlessness and existential grief [50].

Rush and Misajon [51] identified loss of control as a central topic relevant to patients with endometriosis. Young patients in particular reported feelings of frustration regarding educational/job opportunities and intimate relationships [51]. The loss domain is often associated with symptoms of depression [20] that are also quite common among patients with endometriosis; patients with endometriosis show symptoms of depression more often than healthy controls [52] and are more likely to be diagnosed with major depression or other forms of depression over their lifetime [13]. 

The experience and intensity of chronic pain is discussed as a moderating variable for depressive symptoms [53,54], although some of the behaviors listed as typical for the Loss domain, such as worrying and being biased toward negatively valenced information, might also influence the psychological burden of living with endometriosis. In their study, Van Aken et al. [25] found that pain catastrophizing independently influences health-related quality of life (HRQoL), even when pain intensity was included in their regression model. When looking at sexual stress, negative metacognitive beliefs seem to play an even larger role. In the cross-sectional study of Zarbo et al. [55], negative metacognitive beliefs predicted sexual distress in hierarchical logistic regression, while dyspareunia and chronic pain did not. Their findings provide support for the presumption that cognitive processes, such as rumination and metacognitive beliefs, have an additional, independent effect on psychological symptom severity. Donatti et al. [56] identified a solution-oriented focus on clear-cut problems instead of catastrophizing as a successful coping strategy associated with decreased symptoms of depression. The cognitive restructuring of unhelpful thoughts was identified as another helpful coping strategy by González-Echevarría et al. [57], as it was associated with higher HRQoL. Facchin et al. [58] highlight the need for actively restoring continuity in living with endometriosis to overcome a sense of disruption and loss. Hållstam et al. [50] stress the importance of professional support and acknowledgement throughout the process of grief, so that a sense of coherence and the experience of a purpose in life can be re-established. 

## 3. Systems for Social Processes

The domain Systems for Social Processes subsumes all reactions to interpersonal events and interactions regarding different social contexts. It contains the four subconstructs: Affiliation and Attachment, Social Communication, Perception and Understanding of Self, and Perception and Understanding of Others. Affiliation and Attachment, as well as Perception and Understanding of Self, were included as relevant in the context of psychosocial functioning in those with endometriosis.

### 3.1. Affiliation and Attachment

The Affiliation and Attachment subconstruct describes the processes for friendly social approach and bonding. Affiliation, as social approach behavior and engagement in positive social interactions, can result in attachment, which is selective affiliation. Attention to social cues, as well as social learning and memory, are required to engage in affiliation and attachment. 

The experience of social affiliation, closeness, and forming attachment are fundamental human needs and, oftentimes, preconditions to psychological well-being [59,60]. As endometriosis symptoms can interfere with work, social activities, and hobbies [47,61], patients have less time and fewer opportunities to take part in positive social interaction. They are, therefore, less able to experience positive reinforcement through positive social interaction [62], which might contribute to the risk of developing depressive symptoms. The diagnosis of endometriosis comes with many barriers to societal participation; some patients describe the need for the spontaneous cancellation of plans due to symptoms such as pain, irregular bleeding [63], or fatigue [64], missing out on family events, and fear of letting other people down [65]. Further social withdrawal seems to be a consequence of not feeling understood by friends and family members [66], resulting in increased feelings of loneliness and isolation [65,67].

The extensive effects of endometriosis on patients’ day-to-day life becomes even more apparent when considering its influence on intimate romantic relationships and family life. Other than not participating in as many social occasions to meet potential new partners, patients describe feelings of shame and fear with regard to dating [63], because they anticipate being a burden to potential new partners. They often find it particularly difficult to disclose how dyspareunia and vaginal bleeding affect their experience of penetrative sex and physical intimacy [63]. Some patients even prefer the silent endurance of pain during and after intercourse over engaging in a conversation with their partner [68].

In established romantic relationships, endometriosis can have a tremendous effect on relationship dynamics [46,69,70] and requires individually aligned coping strategies, as symptom severity in endometriosis, marital satisfaction, and sexual satisfaction are each associated with the other [71,72]. Many patients experience sexual distress, since they are nine times more likely to experience dyspareunia than healthy controls. Loss of satisfying sex life can occur due to pain [73], bleeding [63], and other kinds of impairments in sexual functioning [7]. In Fritzer et al.’s study [74], patients with endometriosis and dyspareunia reported less sexual intercourse, and disruption or avoidance of it (with 46% of participants stating that their partner’s satisfaction was their main motivation for sexual interactions). Pluchino et al. [72] underline the role of other determinants of sexual health apart from dyspareunia. Cognitive coping strategies, such as catastrophizing and a partner’s negative reaction to sexual pain, might aggravate distress. Hence, it is not surprising that Van Niekerk et al. [75] report an association between vulvar/clitoral pain and lower quality of life in their cross-sectional study. Many couples report feeling left alone by health practitioners regarding their sex life [46].

Reduced fertility or infertility is another severe burden for those trying to conceive [58], further adding to the strain on sexual health. In the sample of Fritzer et al. [74], 30% of participants named wanting to conceive as the main motivation for penetrative sex. In a qualitative study by Márki et al. [76], some patients even recall losing a previous partner due to sexual distress or having to undergo strenuous fertility treatment. Infertility, or the risk thereof, is also perceived as a threat to female identity by some patients [71]. Believing that childless women were of less value than mothers was associated with lower mental health and self-esteem for patients in Facchin et al. [77]. These effects could then, again, affect relationship dynamics negatively [71]. Additionally, patients with children oftentimes report the negative impact of endometriosis on domestic duties and childcare [78,79]. Their concerns include not being able to play with them [12] and their illness limiting family activities [50]. One participant in a qualitative study by Jones et al. [12] described worrying whether her daughter will receive adequate care for her own endometriosis symptoms.

On the oher hand, helpful social support in romantic and other personal relationships can play an important positive role in coping with endometriosis [65]. Márki et al. [76] highlight the need for adequate, reliable information enabling both patients and partners to engage in useful coping strategies. Dyadic coping in couples living with chronic illness is associated with better physical health, well-being, and overall relationship satisfaction [80]. In couples dealing with endometriosis, McKay et al. [81] also discovered a link between higher levels of perceived emotional intimacy and the relationship satisfaction of both partners. Overcoming the joint struggle of living with endometriosis could even serve as an opportunity for mutual growth, creating a stable, lasting relationship [46]. 

In the last decades, another means of social support for people with endometriosis in the form of online communities has emerged [75]. Most endometriosis patients are open to finding information and sharing experiences online, with higher trust in official endometriosis sites [82]. Online communities for people with physical disabilities have been proven effective in offering social support and advice [83]. Thiel et al. [84] even suggest utilizing this interest in online platforms to provide more well-founded information to avoid nocebo-effects in treatment. Online platforms and communities seem to be a promising approach for additional support. 

### 3.2. Perception and Understanding of Self

The subconstruct Perception and Understanding of Self includes the two subconstructs of agency and self-knowledge. It describes processes and representations for assessing one’s own internal states and traits, and for supporting self-awareness, self-monitoring, and self-knowledge. 

Many patients with endometriosis describe changes in or loss of agency, especially when in acute pain [85]. In their qualitative study, Bullo et al. [86] found some patients to share a perception of pain as the loss of agency to an externalized attacker. These findings are supported by a study in which endometriosis patients wrote narratives about their life with the disease, finding 68% of the sample to feel powerless, at least to some extent [87]. This feeling of missing agency was significantly positively correlated with depressive symptoms and neuroticism, while being negatively correlated with life satisfaction. Other patients even describe disconnection from their thoughts and losing their sense of self, largely due to the overwhelming intensity of pain and becoming paralyzed during its peaks [85], illustrating that severe pain can, in fact, reduce agency.

With regard to the evaluation of their own body, endometriosis patients show higher levels of body image concerns [88,89]. In Sayer-Jones et al.’s qualitative study [90], patients reported experiencing a sense of betrayal from their own body or compared their body to a prison. Another participant stated that postoperative scarring made her feel unattractive [90]. Geller et al. [91] found that body image and self-criticism moderated differences in depression and anxiety levels between patients. In turn, Van Niekerk et al. [35] discovered an association of body compassion with higher HRQoL. Falconer [92] criticizes the methodological issues of existing research on body image in endometriosis, pointing out the need for a standardized assessment of satisfaction with body image and body image concerns.

Concerning the appropriate perception of their competences, skills, beliefs, and desires, patients with endometriosis might be vulnerable to developing deficits in this area. In particular, if their self-confidence is impacted, they might suffer from it even more than healthy controls. Higher rates of self-criticism mediated differences in the symptoms of depression in the study of Geller et al. [91]. Marschall et al. [87] also found that some patients’ illness narratives, which centered around negative self-change, were associated with more symptoms of depression in comparison to narratives centered around less negative self-change and communion. González-Echevarría et al. [57] report on self-criticism as a negative coping strategy associated with lower HRQoL.

In accordance with the cognitive model of depression, negative self-evaluation might constitute a risk factor for the development of depressive symptoms [93]. Cause–effect relations of negative self-evaluation and depressive symptoms remain unclear within the context of the psychological symptoms associated with endometriosis. Nevertheless, these findings highlight the need for mind–body interventions that target psychological symptoms and hopefully improve HRQoL. 

## 4. Discussion

In our review, previous findings on psychosocial functioning in endometriosis were restructured, utilizing the RDoC framework. The domains Negative Valence Systems and Social Processes were explored, aiming to illustrate the transdiagnostic interrelations in symptom perpetuation.

It becomes apparent that endometriosis can have far-reaching psychological consequences. Endometriosis is stress-associated, with HPA dysregulation supporting chronic inflammation and pain chronification. A high physical and mental load, combined with loss of resources, can result in higher levels of stress and vigilance toward the worsening of symptoms. People who experienced childhood abuse are more likely to develop endometriosis later in life due to their already dysregulated stress response. Fearful symptom monitoring, in turn, can cause a worsening of symptoms, lower self-esteem, rumination, and negative meta-cognitive beliefs affecting many areas of life; endometriosis can interfere with work, social activities, hobbies, and relationship/family life and family planning. Therefore, patients might withdraw from social encounters and activities, which can lead to a complex cycle of lowered well-being, more social isolation, less positive reinforcement, and feelings of loneliness and despair. Romantic and sexual relationships can be especially affected by endometriosis symptoms. Taken together, endometriosis is a disease that can affect all areas of life. It creates self-sustaining mechanisms of disease progression that interact with each other, partially leading to symptoms that persist even after the extensive removal of endometrial tissue.

These findings support the call of other authors (e.g., [94,95]) who demand that endometriosis be widely recognized as a systemic disease. An early diagnosis and an early start of not just medical therapy is vital in order to avoid the emergence of the described connections and, thus, to maintain well-being and quality of life. 

### 4.1. Wider Implications for Treatment and Future Research

There is a need for additional multimodal therapy, focusing on broader stress and pain processes in endometriosis, to reverse the many pathogenetic mechanisms that play a role in the progression of psychological symptoms.

Psychotherapy for patients with endometriosis would have to acknowledge and validate all these challenges, while at the same time providing profound tools for mastering them and re-establishing a sense of coherence [50]. Pilot studies show a positive effect of different mind–body interventions [96,97,98,99], which matches study findings demonstrating that patients often wish for more holistic care [100]. In particular, studies including elements from cognitive behavioral therapy (CBT) could significantly improve quality of life in patients [101]. CBT has been proven as effective in the treatment of chronic pain [102], depression [103], and chronic stress [104], making it a promising candidate for improving HRQoL in endometriosis. CBT allows for the restructuring of automatic thoughts, as well as acceptance of events in the outer world and of internal experiences (e.g., uncomfortable thoughts and emotions) [105]. Patients could probably benefit from these strategies when dealing with symptoms such as rumination, high levels of perceived stress, and fear of movement. To the authors’ knowledge, no RCT study has yet examined the effects of CBT interventions for patients with endometriosis, although some study protocols have been published [106,107]. Some potentially useful treatment elements could be psychoeducation, pacing daily activities and movement, navigating the workplace, relationship and sexual therapy, managing fertility treatment, and/or grieving infertility. The feeling of loss of agency and alienation from one’s own body could be specifically tackled, e.g., with self-compassion-based interventions [108]. Research on psychosexual interventions in endometriosis patients is still scarce, but it is promising for pain symptoms and sexual functioning [95]. With regard to any kind of additional treatment, individualized therapy approaches are necessary based on the individual symptomatic profile. Every patient is affected by endometriosis in a possibly unique way, although the various symptoms stem from quite distinct somatic grounds. 

There is still a knowledge gap regarding the etiology, pathogenesis, pain chronification, subfertility mechanisms, and cause–effect relationships in the context of endometriosis. The impact of endometriosis on not just romantic relationships, but on the family system and child development, is still to be explored in more detail. Furthermore, the psychological long-term effects of different kinds of treatment need to be further investigated. Research on mind–body interventions for patients with endometriosis has only emerged over the last two decades. In order to generate better screening tools, to potentially develop preventive programs, and to improve any kind of treatment, we need more research, especially longitudinal cohort studies. Research on endometriosis could benefit from consistent use of validated outcome measures for not only pain, but HRQoL. Bourdel et al. [109] discuss the strengths and weaknesses of commonly used outcome measures and recommend the use of the SF-36 [110] and the EHP-30 [111]. The EHP-30 is an endometriosis-specific HRQoL questionnaire that was developed from interviews with patients [111]. The EHP-30 + 23 consists of a core questionnaire with five subscales and six modular components, covering different areas of life possibly relevant to patients (e.g., work life, infertility). It is sensitive to change [112] and relatively easy to administer [109].

Hudson [113] points out that endometriosis has suffered from a lack of recognition and research funding for decades, even though its significant impact on patients is widely recognized already. Additionally, not all individuals affected by endometriosis receive the same attention in the medical care system. Gender, race, culture, and class play important roles in the quest for finding the right diagnosis and receiving adequate treatment [114,115]. The influence of these social intersections should be acknowledged by researchers and practitioners and carefully kept in mind when designing studies, analyzing data, and treating patients. Especially in endometriosis, gender is a crucial factor; the lack of recognition and funding might partially be an indirect result of academic research being shaped by male researchers not taking a so-called women’s disease seriously [113]. At the same time, whenever endometriosis is reduced to a women’s disease, affected individuals who do not identify as female (e.g., trans men or nonbinary individuals) are excluded. They may, on the one hand, experience gender dysphoria whenever confronted with their so-called female disease [116]. On the other hand, the process of transitioning and undergoing testosterone treatment and/or hysterectomy sometimes interferes with endometriosis symptoms and treatment [117]. More research on how to provide trans, intersex, and nonbinary individuals with safe and adequate health care free of discrimination is needed. 

### 4.2. Limitations

Although we provide a detailed and comprehensive insight into the complex mechanisms creating psychosocial burden, our review is not systematic and does not offer the same accuracy as could have been reached with a meta-analytic approach. However, meta-analyses and systematic reviews on endometriosis often suffer from lack of methodological quality in original studies and call for studies with clear, replicable study designs and better reliability (e.g., [101,118]). Instead of conducting a systematic literature review, the linking of existing evidence on the psychosocial burden of endometriosis across diagnoses seemed to be of value. Therefore, the RDoC structure, with its transdiagnostic dimensional perspective, provided a helpful framework for this review. Its neurobiological foundations and the idea of mapping psychopathological phenomena with distinct (neuro)physiological circuits, molecules, and genes might be perceived as reductionistic [119] when used with the intent of forming conclusive models of every interrelation between units and domains of analysis. Conceptualizing psychiatric disorders as merely brain disorders in the sense of one-to-one correspondence is, indeed, not doing justice to the complexity and interpersonal heterogeneity of human experience [119]. However, it might be helpful to keep in mind that RDoC were primarily designed as a constantly evolving research tool to inspire constructive dialogue about integrating neurobiological findings into the understanding of mental illness [18]. They are meant to be an additional framework to the International Classification of Diseases (ICD) and Diagnostic and Statistical Manual of Mental Disorders (DSM) [120], not a superior framework to replace them. In the context of endometriosis, where relations between the different factors contributing to progression of the disease itself, and to the development of mental burden, remain unclear, they can provide a new perspective supporting the need for psychotherapeutic treatment. The perspective on endometriosis through the RDoC framework can be of additional value for future research.

## 5. Conclusions

In this review, we presented the psychosocial mechanisms within, and as a consequence of, endometriosis, a disease that can, using the RDoC, be defined as systemic rather than only gynecological. We were able to show the different facets of the condition and how it impacts wellbeing and health-related quality of life. It became clear that early diagnosis and adequate, multimodal treatment are vital. Specifically, mind–body interventions, such as psychotherapy to reduce stress and support healthy coping, are needed in addition to medical care. On a societal level, endometriosis needs to be taken more seriously, since it can put such a strain on a patient’s quality of life. Better availability of knowledge will hopefully shorten the time between symptom onset and correct diagnosis and, thus, help to halt the chronification processes early on, as well as generate more public interest, leading to more research and treatment funding.

## Data Availability

Not applicable, as no new data were created.

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
