# Peer review of "Understanding Psychological Symptoms of Endometriosis from a Research Domain Criteria Perspective"

_jcm, 2023, doi:10.3390/jcm12124056_

Round 1

Reviewer 1 Report

Dear Editor,

thank you for inviting me to review the manuscript entitled “Understanding psychological symptoms of endometriosis from a research domain criteria perspective” by Katharina van Stein et al. 

The objective of this narrative review is to assess and to analyze the several psychological implications which may burden the quality of life of women affected by endometriosis. 

Moreover, Authors provide evidence occurred by the Research Domain Criteria project, which suggests a multidimensional approach to mental health and illness potentially related to the disease and considers both the behavioral and neurobiological mechanisms involved.

The abstract is clear and well summarizes the contents of the paper, all sections are well structured, references are updated, the English language is appropriate and well spelled, and only few typing errors were detected in the text. 

There are no tables and figures. Statistical analysis is not required for this type of article.

The topic is very interesting, it fits with the journal scope, and opens to the possibility of discussing on unexplored issues in this field. Conclusions are consistent with the evidence and arguments presented and are supported by the listed citations. 

Therefore, the article deserves publication in the Journal of Clinical Medicine after minor revision.

Comments: 

-       To assess the self-perception of quality of life there are several validated questionnaires, such as the EHP-30, able to provide an objective evaluation of the health status of affected patients. Please, consider this aspect in the Discussion.

-       Patients often resent the effect of long-term medical treatments, due to the changes in their bodies and/or to the prolonged side effects which could occur. Very little evidence is available in this field (see: Piacenti I. et al. “Dienogest versus continuous oral levonorgestrel/EE in patients with endometriosis: what's the best choice?” Gynecol Endocrinol. 2021. doi: 10.1080/09513590.2021.1892632 or Alcalde AM et al. “Assessment of Quality of Life, Sexual Quality of Life, and Pain Symptoms in Deep Infiltrating Endometriosis Patients with or Without Associated Adenomyosis and the Influence of a Flexible Extended Combined Oral Contraceptive Regimen: Results of a Prospective, Observational Study” J Sex Med. 2022. doi: 10.1016/j.jsxm.2021.11.015). Please analyze this aspect in the text. 

-       As reported by the Authors, one of the hot topics in this field is that of infertility, and relevant evidence on its burden was reported. On the other hand, little is known about the quality of life of women with symptomatic endometriosis occurred after pregnancies, which can impact on the relationship with children, especially due to both depressive mood and chronic fatigue. Did Authors find any evidence in regard? 

-       Authors mention the effects of surgery on psychological stress. However, another relevant aspect is that of the body self-perception after surgery, especially if repeated or demolitive (e.g. intestinal resections with placement of permanent stomia, laparotomic surgery with visible scars, …). Please, report the available evidence and discuss this topic.  

The English language is appropriate and overall well spelled, but few typing errors were detected in the text. Please, provide a linguistic check by a native-English speaker.

Reviewer 2 Report

Dear Authors,

The review represents an important scientific contribution to the topic of endometriosis. The combination of results from different fields highlights the complexity of this disesase. The structuring of the results found in the literature using the RDoC framework also helps to present the state of research from a different perspective.

Even though the results are based on an unsystematic review, a short paragraph on the procedure (e.g. databases, keywords, year of publication) should be included. The large amount of literature cited suggests that a research was conducted, even though inclusion and exclusion criteria were not defined. A brief description of the methodology will help to support the scientific statement.
